# Snow Density Retrieval in Quebec Using Space-Borne SMOS Observations

Xiaowen Gao [1,2], Jinmei Pan [1,*], Zhiqing Peng [1,2], Tianjie Zhao [1], Yu Bai [1,2], Jianwei Yang [3], Lingmei Jiang [3], Jiancheng Shi [4] and Letu Husi [2]

[1] State Key Laboratory of Remote Sensing Science, Aerospace Information Research Institute, Chinese Academy of Sciences, Beijing 100094, China
[2] University of Chinese Academy of Sciences, Beijing 100049, China
[3] State Key Laboratory of Remote Sensing Science, Jointly Sponsored by Beijing Normal University and Aerospace Information Research Institute of Chinese Academy of Sciences, Faculty of Geographical Science, Beijing Normal University, Beijing 100875, China
[4] National Space Science Center, Chinese Academy of Sciences, Beijing 100190, China
* Correspondence: panjm@aircas.ac.cn

**Abstract:** Snow density varies spatially, temporally, and vertically within the snowpack and is the key to converting snow depth to snow water equivalent. While previous studies have demonstrated the feasibility of retrieving snow density using a multiple-angle L-band radiometer in theory and in ground-based radiometer experiments, this technique has not yet been applied to satellites. In this study, the snow density was retrieved using the Soil Moisture Ocean Salinity (SMOS) satellite radiometer observations at 43 stations in Quebec, Canada. We used a one-layer snow radiative transfer model and added a $\tau - \omega$ vegetation model over the snow to consider the forest influence. We developed an objective method to estimate the forest parameters ($\tau$, $\omega$) and soil roughness ($S_D$) from SMOS measurements during the snow-free period and applied them to estimate snow density. Prior knowledge of soil permittivity was used in the entire process, which was calculated from the Global Land Data Assimilation System (GLDAS) soil simulations using a frozen soil dielectric model. Results showed that the retrieved snow density had an overall root-mean-squared error (RMSE) of 83 kg/m$^3$ for all stations, with a mean bias of 9.4 kg/m$^3$. The RMSE can be further reduced if an artificial tuning of three predetermined parameters ($\tau$, $\omega$, and $S_D$) is allowed to reduce systematic biases at some stations. The remote sensing retrieved snow density outperforms the reanalysis snow density from GLDAS in terms of bias and temporal variation characteristics.

**Keywords:** snow density; SMOS; multiple-angle; passive microwave remote sensing

## 1. Introduction

Snow cover plays a critical role in terrestrial hydrological, climatological, and ecological processes. It influences the energy balance on the land surface, based on its high albedo and low thermal conductivity [1]. The measurement of snow water equivalent (SWE) is important to understand the timing and magnitude of snowmelt runoff [2]. However, when there have been multiple continental or large-scale snow depth remote sensing products, the corresponding snow density product required to convert snow depth to SWE is still limited. Snow density, SWE, and snow pressure are not as widely measured as snow depth at stations. When the passive microwave remote sensing technique at the Ku- to Ka-band was used to retrieve the snow depth for more than four decades [3], observations at these frequencies were rarely used to estimate snow density. Instead, after the snow depth is retrieved, the error in snow density auxiliary information becomes an important source of SWE uncertainty [4]. At a high frequency, snow volume scattering cannot be neglected. The influences of the snow depth and snow grain size on the brightness temperature (T$_B$) are much higher than that of snow density. The coherency between scattered waves from

snow particles [5] results in a sophisticated relationship between $T_B$ and snow density, which is differently understood by different snow radiative transfer models [6]. Therefore, to estimate snow density, the lower-frequency channel is better suited for its retrieval as it is more sensitive to snow refraction rather than snow volume scattering [7,8].

Snow density varies spatially, temporally, and vertically, influenced by the snow compaction rate and snow compaction time [9–11]. The use of a fixed snow density (for example, 240 kg/m$^3$) will result in an overestimated SWE in the early snow season and an underestimated SWE in the late snow season [12–14]. The CCI+ Version2 (CCIv2) SWE product [13] utilized time-varying snow density functions according to snow classes [15]. The empirical method that calculates the snow density as a regression function of the snow depth, day of year, and snow classes [16] is unable to capture the spatial and inter-annual variability [17]. In a warming climate, the direct observation of snow density can be of great value to detect the increased occurrence of rain-on-snow events.

In the field, the most commonly used method to measure snow density is to weigh the snow in a container of a fixed volume [18]. Dielectric permittivity measurement [19], micro-computed tomography [20,21], and neutron-scattering probes [22] can also be used to measure snow density. After being inter-calibrated against the gravimetric measurements, these techniques are either fast in the field or of fine vertical resolution [23–25]. Snow density can also be estimated by L-, C-, and X-band synthetic aperture radar (SAR) data. The study in [26] utilized the change in surface scattering caused by snow, and estimated the snow density using a parameterized backscattering model simplified from the Integral Equation Model (IEM) [27]. In recent years, with the development of full-polarized SAR, surface [28,29] and volume scattering [30,31] components were extracted from different signal decomposition methods to establish empirical or physical relationships with snow density.

After the launch of the Soil Moisture Ocean Salinity (SMOS) [32] satellite for soil moisture observation purposes in November 2009, the multiple-angle L-band radiometer became an additional type of sensor applicable for snow density estimation. The revisit frequency of SMOS is two to three days, which makes it suitable for studying temporal snow density variation and supporting daily SWE products. Preliminary studies on the physical [8] and the practical feasibility [7] of snow density retrieval using ground-based radiometer observations have been conducted. It was found that the existence of snow on the bare soil changes both the propagation angle of microwaves inside the snow medium and the refraction at the air–snow and snow–soil boundaries. With the increase in snow density, the propagation angle becomes steeper, the emissivity at the horizontal polarization increases, and the emissivity at the vertical polarization increases at small incidence angles and decreases at large incidence angles. Because of the large contrast between the L-band wavelength and snow grain size, volume scattering generated by snow particles can be neglected [7,8,26]. This means that the computational cost to calculate snow $T_B$ and retrieve the snow density can be largely reduced. Using the simplified snow radiative transfer model, the studies in [7,33,34] successfully estimated the snow density in tower-based radiometric experiments. However, if the snow density retrieval algorithm is moved from ground-based to space-borne instruments, it will be challenged by stronger observation noise, emission from other land surface types, and the Radio Frequency Interference (RFI) from human activities. Whether the snow density is retrievable becomes uncertain.

Therefore, the scientific goal of this article is to retrieve the snow density from SMOS satellite observations, according to the theory in [7,8]. The study area is Quebec, Canada, determined by the availability of ground measurements and the appropriate terrain condition. Forests in Quebec should be considered using space-borne observations in theory. To minimize the impact of forests during snow density retrieval, the τ-ω vegetation model was utilized. The prior knowledge of soil permittivity was introduced and calculated from the Global Land Data Assimilation System (GLDAS) simulated soil temperature and total soil water content [35,36] using a frozen soil dielectric model [37]. We also developed a method based on objective optimization to determine $\omega$, $\tau$, and soil roughness ($S_D$) using

the SMOS observations in the snow-free period and applied them subsequently to estimate the snow density in the snow season. With the predetermined parameters and the prior knowledge, the ill-posed problem of snow density retrieval was greatly relieved.

This paper is organized as follows. Section 2 introduces the study area and datasets utilized. Section 3 describes the forward emission model and the retrieval algorithm. Section 4 shows the snow density retrieval results. Sections 5 and 6 are the Discussion and Conclusions, respectively.

## 2. Study Area and Datasets

### 2.1. Study Area

The study area is located in Quebec, Eastern Canada (45°N–63°N, 57°W–80°W), with elevation above sea level from −3 to 1081 m (Figure 1). Characterized by cool temperatures in summer and abundant snowfall in winter, this area includes eight different terrestrial ecozones [38]. Snow in this region has large spatial variability, with a snow cover duration ranging from 120 days in Southern Quebec to 240 days in Northern Quebec, and an annual maximum SWE from less than 100 mm at low altitudes to more than 300 mm at high altitudes [39,40].

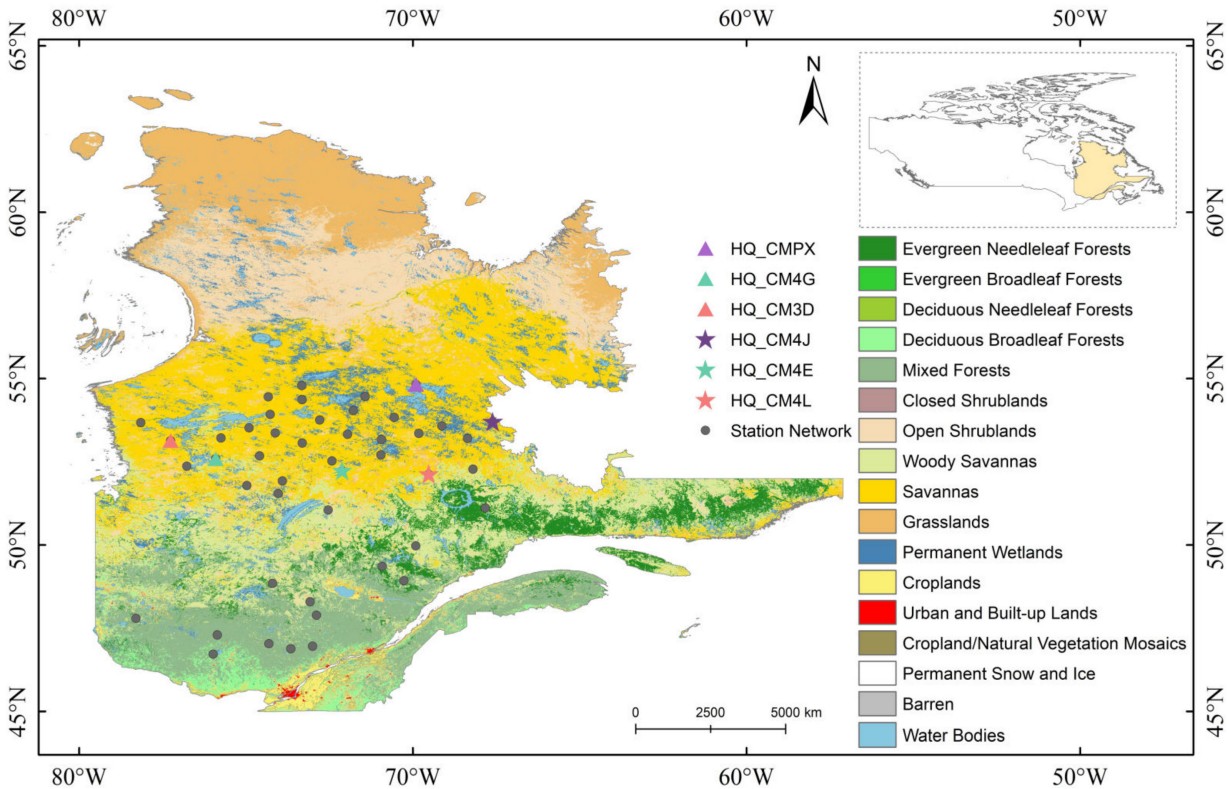

**Figure 1.** Spatial distribution of stations in Quebec, Canada. International Geosphere-Biosphere Programme (IGBP) land surface types are represented in colors according to the MODIS-MCD12Q1 product. Stations marked with stars and triangles are stations with good and bad retrieval performances, respectively, described in detail in Section 4.

### 2.2. Datasets

#### 2.2.1. The SMOS Brightness Temperature Dataset

The SMOS mission launched by the European Space Agency [32] provides L-band $T_B$ at multiple observation angles and dual polarizations. The SMOS Level 1C (L1C) product is provided in the ISEA-4H9 grid (icosahedral Snyder equal area earth fixed) with a spatial resolution of 43 km on average [41]. However, the observations in some areas are affected by the RFI. We employed a two-step regression approach, developed in [42], to smooth the

angular dependence of the SMOS L1C $T_B$ product. The method can largely remove outliers and reduce the observation uncertainty, which was validated in [43]. In the post-processed product, the SMOS $T_B$ was resampled to fixed incidence angles at 40° and from 2.5° to 62.5° with an interval of 5°. Observations at all angles from ascending orbits with an overpass time of 6:00 AM were used for snow density retrieval. The study period was from October 2019 to June 2020.

### 2.2.2. In-Situ Snow Measurements

The in-situ snow measurements come from the Canadian historical Snow Water Equivalent dataset (CanSWEv3.0) [44,45], which contains both the snow depth and snow water equivalent at the same station. The entire dataset has measurements from 2832 stations in the 1928–2021 period. In our study area and period, there are 43 stations with elevation < 1000 m. According to the International Geosphere-Biosphere Programme (IGBP) classification acquired from the MODIS MCD12Q1 product [46], stations in the south are located in highly forest-covered areas, including evergreen needleleaf forests, mixed forests, and woody savannas, whereas the remaining are mostly located in the savanna region. Stations falling within grids with a dominant IGBP type of water bodies were removed.

### 2.2.3. Other Auxiliary Datasets

To calculate the ground permittivity, we introduced the simulated total soil water content and soil physical temperature at 0–10 cm from the Global Land Data Assimilation System (GLDAS) Noah model product [35,36]. A soil dielectric model was used to calculate the frozen soil permittivity [37], and the required soil texture (sand, silt, and clay contents) and soil bulk density were extracted from the Harmonized World Soil Database [47]. In the $T_B$ calculation, the vegetation temperature was considered the same as the air temperature from GLDAS, whereas the snow temperature was calculated as the mean of the GLDAS soil temperature and air temperature. The snow temperature has a very small influence on the imaginary part of snow permittivity. To calculate the influence of the forest canopy, we introduced the forest cover fraction from the MODIS MOD44B product [48].

## 3. Methods

### 3.1. Forward Emission Model

The radiative transfer model used to describe the emission of the soil–snow–vegetation system is an empirical rough soil reflectivity model [49], coupled with a simplified snow emission model neglecting absorption and scattering coefficients [7] and a $\tau - \omega$ vegetation model [50]. For an SMOS grid with a forest cover fraction of $F_C$, the measured $T_B$ can be written as in Equation (1), which includes an open snow component (of areal fraction of $1 - F_C$) and a forest-covered snow component (of areal fraction of $F_C$):

$$T_B^p = T_{B,f}^p F_C + T_{B,S}^p (1 - F_C)$$ (1)

where $T_B^p$ is the satellite-observed $T_B$ at polarization $p$ ($p$ = H or V), $T_{B,S}^p$ is the open snow $T_B$, and $T_{B,f}^p$ is the snow-emitted $T_B$ observed above the forest canopy. The SMOS grid is considered fully snow-covered.

$T_{B,f}^p$ is expressed as a three-component model as [50]:

$$T_{B,f}^p = T_{B,S}^p \gamma + T_C (1 - \omega)(1 - \gamma) + T_C (1 - \omega)(1 - \gamma) r^p \gamma$$ (2)

where $T_C$ is the vegetation physical temperature. $\omega$ is the effective scattering albedo accounting for the volume scattering of the canopy [51], and $\gamma$ is the vegetation attenuation factor. The first component is the snow emission attenuated by the vegetation canopy, the second component is the thermal emission from the forest propagated upward, and the third is the thermal emission from the forest propagated downward, reflected by the snow and attenuated by the vegetation canopy. $\omega$ and $\gamma$ are assumed independent of

polarization in this study. $\gamma$ is expressed as $\gamma = \exp(-\tau \sec\theta)$, where $\tau$ is the vegetation optical depth (VOD) and $\theta$ is the observation angle. $r^p$ is the effective surface reflectivity of the snow–soil system.

Due to the low-loss characteristic of snow cover in the L-band [52], the simulation of $T^p_{B,S}$ in this study utilized the simplified snow emission model neglecting snow absorption and volume scattering, as [7]

$$T^p_{B,S} = a^p_G \, T_G + a^p_S \, T_S + a^p_{sky} \, T_{sky} \tag{3}$$

where $T_G$, $T_S$, and $T_{sky}$ are the soil physical temperature, snow physical temperature, and downwelling sky $T_B$, respectively. If the snow is homogeneous, the Kirchhoff coefficients $a^p_G$, $a^p_S$, and $a^p_{sky}$ can be expressed as

$$
\begin{aligned}
a^p_G &= \frac{(1 - s^p_G)\,(1 - s^p_S)\,t_S}{1 + r^2_S\, s^p_G\, s^p_S - r_S\,(s^p_G + s^p_S) - t^2_S\, s^p_G\, s^p_S} \\
a^p_S &= \frac{(1 - s^p_S)\,(1 - r_S - t_S)\,(1 - r_S s^p_G + t_S s^p_G)}{(1 - r_S s^p_G)\,(1 - r_S s^p_S) - t^2_S\, s^p_G\, s^p_S} \\
a^p_{sky} &= \frac{(1 - r_S s^p_G)\,(s^p_S + r_S\,(1 - 2s^p_S)) + s^p_G\,(1 - 2s^p_S)\,t^2_S}{(1 - r_S s^p_G)\,(1 - r_S s^p_S) - t^2_S\, s^p_G\, s^p_S}
\end{aligned} \tag{4}
$$

where $s^p_G$ is the snow–soil interface reflectivity, $s^p_S$ is the specular air–snow interface reflectivity, and $r_S$ and $t_S$ are the reflectivity and transmissivity of the snow layer, respectively.

When there is no absorption, no volume scattering assumptions are made for snow, $t_S = 1$ and $r_S = 0$. The Kirchhoff coefficients can be simplified to

$$a^p_G = \frac{(1 - s^p_G)\,(1 - s^p_S)}{1 - s^p_G\, s^p_S} \quad a^p_S = 0 \quad a^p_{sky} = 1 - a^p_G \tag{5}$$

where $a^p_G$ can be used as $1 - r^p$ in Equation (2).

We evaluated the reasonability of neglecting absorption and volume scattering at the L-band using the Microwave Emission Model of Layered Snowpacks (MEMLS) based on the Improved Born Approximation (IBA) [53,54], and found that the sensitivity of the snow-emitted $T_B$ to snow depth is only 0.018–0.072 K per m, using an exponential correlation length of 0.18 mm.

To calculate $s^p_G$, a semi-empirical QHN model [55,56] with frequent-independent coefficients [49] was used (Equations (6) and (7)). It allows the use of one parameter, the standard deviation of height ($S_D$), to model the rough soil surface reflectivity, which is easier to fit than models using two parameters:

$$s^p_G = \left((1 - Q_R)\, s^{p*}_G + Q_R\, s^{q*}_G\right) \exp\left(-H_R \cos^{N_{Rp}} \theta\right) \tag{6}$$

$$H_R = \left(\frac{a_1 S_D}{a_2 S_D + a_3}\right)^6 \tag{7}$$

where $s^{p*}_G$ and $s^{q*}_G$ are calculated from the soil permittivity using Fresnel equations [57] at polarization $p$ and $q$ (e.g., $q$ = V if $p$ = H and vice versa). The soil dielectric model developed in [37] was used to calculate the soil permittivity. The remaining parameters were optimized as $Q_R$ = 0.075, $N_{RV}$ = 1.503, $N_{RH}$ = 0.131, and $H_R$ is a function of $S_D$, with coefficients as $a_1$ = 0.887, $a_2$ = 0.796, $a_3$ = 3.517 [49].

### 3.2. Retrieval of Predetermined Parameters ($\tau$, $\omega$, $S_D$) in Snow-Free Period

To reduce the difficulty in snow density estimation, three unknown parameters ($\tau$, $\omega$, $S_D$) were predetermined based on SMOS observations in the snow-free period. We assumed that the values of the three parameters remain constant over time and are independent of

polarization. Using the soil permittivity from GLDAS, we calculated the error between the simulated and the observed SMOS $T_B$ during the snow-free period, using different combinations of $\tau$, $\omega$, and $S_D$.

$$T_{B,\,error}(\tau,\,\omega,\,S_D) = \sum_{t=1}^{t_n} \sum_{\theta_k=\theta}^{\theta_n} \sum_{p=H,V} \left( T_{B,\,obs}^p(\theta_k,t) - T_{B,\,mod}^p(\theta_k,\,t,\,\tau,\,\omega,\,S_D) \right)^2 \quad (8)$$

$T_{B,\,error}$ was calculated from days within 2 weeks before the onset or after the end of the snow season. The searching ranges of $\tau$, $\omega$, and $S_D$ were $\tau$ from 0 to 0.5 with a step of 0.01, $\omega$ from 0 to 0.4 with a step of 0.01, and $S_D$ from 0 to 100 mm with a step of 1 mm. It was found that a few combinations of $\tau$, $\omega$, and $S_D$ gave similarly small $T_{B,\,error}$. We extracted the first 0.1% smallest $T_{B,\,error}$ and found the corresponding $\tau$, $\omega$, and $S_D$ combinations (see Figure 2 as an example for the HQ-CM4L station). Later, the values of these $\tau$, $\omega$, and $S_D$ combinations were averaged, and the single combination closest to the averaged value was chosen as the final estimate.

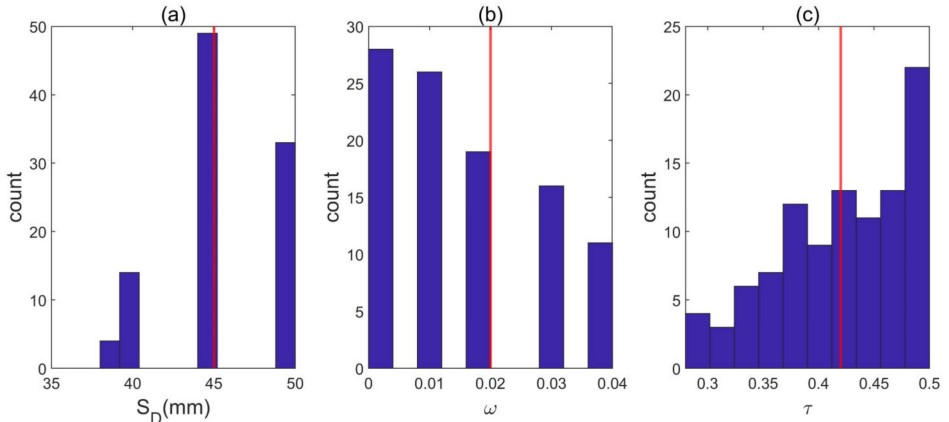

**Figure 2.** The histogram of (**a**) soil–snow interface roughness $S_D$, (**b**) canopy single scattering albedo $\omega$, and (**c**) transmissivity $\tau$ determined by SMOS-observed $T_B$ during the snow-free period at station HQ-CM4L in Quebec, Canada. The red vertical lines are the final estimations of $\tau$, $\omega$, and $S_D$.

### 3.3. Retrieval of Snow Density

After determining the three unknown parameters, it is possible to calculate soil and vegetation emissions. Subsequently, the snow density can be retrieved using a cost function as

$$CF(\rho_S) = \sum_{\theta_k=\theta}^{\theta_n} \sum_{p=H,V} \left( T_{B,\,obs}^p(\theta_k) - T_{B,\,mod}^p(\theta_k,\,\rho_S) \right)^2 \quad (9)$$

The searching range for snow density ($\rho_S$) was 50 to 500 kg/m³ with a step of 1 kg/m³. $\rho_S$ with the smallest $CF(\rho_S)$ will be considered as the retrieved snow density. Figure 3 is the flow chart for the entire snow density retrieval process.

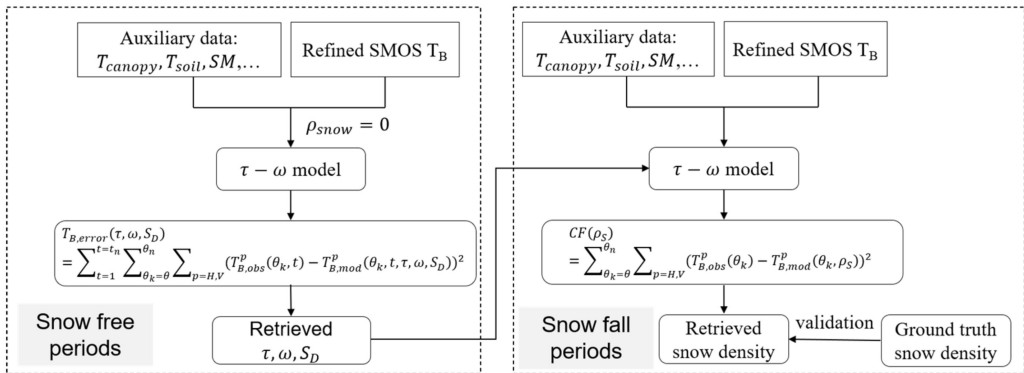

**Figure 3.** Flow chart for snow density retrieval.

### 3.4. Objective Postprocessing Method to Reduce Retrieval Uncertainty

Snow density retrieval is sensitive to the predetermined parameters $\tau$, $\omega$, and $S_D$. In our study, we found that using SMOS measurements from different periods to fit $\tau$, $\omega$, and $S_D$ can lead to different results. Better performance can be achieved by using SMOS measurements before or after the snow season, which varies by station. This is due to changes in bias in the simulated snow-free $T_B$ using the same GLDAS dataset and predetermined parameters, with the impacting factors unknown. To reduce uncertainty and ensure objectivity, we generated three sets of $\tau$, $\omega$, and $S_D$ fitted from before, after, and before and after the snow season, and we generated three sets of snow density time series as candidates. Abnormal snow density time series can be recognized by counting the occurrence of abnormal daily snow density values reaching the 50 or 500 kg/m$^3$ searching boundary limit. The remaining snow density time series will be averaged to obtain the final retrieval result.

## 4. Results

### 4.1. Performance of Multiple-Angle Brightness Temperature Simulation

To examine the behavior of the forward model in matching the SMOS observations, Figure 4 shows examples of simulated and observed $T_B$ for one station during the snow density retrieval. It shows that for the HQ-CM4E station, the simulated $T_B$ matched well with the SMOS-observed $T_B$, except for large incidence angles at horizontal polarization and unstable snow conditions (Figure 4a). On 29 November 2019 (Figure 4a), the relatively low-biased $T_B$ at small angles and the complex snow condition during this season resulted in an underestimation of snow density. However, in Figure 4b–f, the errors between the retrieved snow density and in-situ measurements were within 45 kg/m$^3$. For example, on 6 December 2019, the estimated snow density was 155 kg/m$^3$ with an error of 3.5 kg/m$^3$, and on 16 February 2020, the estimated snow density was 206 kg/m$^3$ with an error of 5.9 kg/m$^3$. On 16 February 2020 (Figure 4d), the increased snow density compared to previous months was retrieved from an increased $T_B$ in small incidence angles, a decreased $T_B$ in large angles at vertical polarization, and an increased $T_B$ at horizontal polarization. On 8 April 2020 (Figure 4f), the model-simulated soil permittivity and the retrieved snow density significantly increased compared to previous examples, which may have been caused by melt–refreeze events in this season.

### 4.2. Performance of Snow Density Retrieval at Example Stations

Figures 5 and 6 show the time series of retrieved snow density using the retrieval method described in Sections 3.2 and 3.3 at six different stations, and we compared them with the CanSWE ground measurements and reanalysis snow density from GLDAS. The GLDAS product provides the snow depth and snow water equivalent, which allows us to calculate the average snow density along the snow profile. Among 43 stations, some have better retrieval performance than others. Therefore, we have selected and presented three examples with good performance in Figure 5 and three examples with poor performance in Figure 6.

As shown in Figure 5, the retrieved snow density captured the increasing trend of the observed snow density from 100~200 kg/m$^3$ in the early and mid- snow season to 300~400 kg/m$^3$ in the late snow season. The root-mean-squared error (RMSE) varies from 52.7 kg/m$^3$ to 67.9 kg/m$^3$ for different stations. The mean bias is smaller than 25 kg/m$^3$, which is approximately 10% of the mean snow density. If we only consider the retrievals during relatively stable snow conditions (December to March), the RMSE can be reduced to 36.7~42.1 kg/m$^3$, but the changes in R and bias are complex, because the snow density variation range is reduced. In the late snow season, the remote sensing retrieved snow density is noisier than the in-situ measurements. However, in the early and mid-snow season, the GLDAS reanalysis dataset underestimates the snow density compared to the measurements. Therefore, the SMOS-retrieved snow density outperforms the GLDAS reanalysis in the time series variation characteristic. In the late snow season after April, the

snow density increases due to melt–refreeze events and snow compaction, and it decreases due to new snowfalls. The observed, SMOS-retrieved, and GLDAS snow density fluctuate at the same pace but with different magnitudes. The shape of the SMOS-retrieved snow density is closer to that of GLDAS, whereas the variation in the observed snow density is smaller in general.

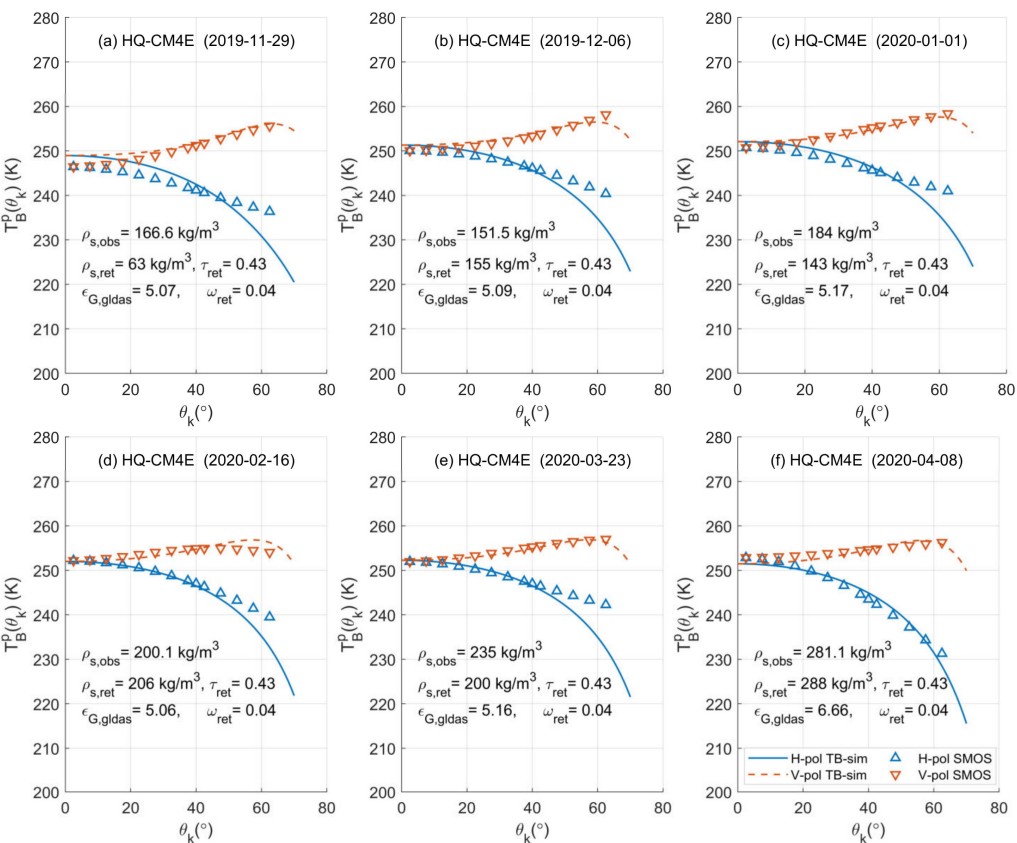

**Figure 4.** Examples of SMOS-observed $T_B$ (triangles) versus the forward-model-simulated $T_B$ (lines) to fit the observations. $\tau_{ret}$ and $\omega_{ret}$ are the retrieved snow density, $\tau$, and $\omega$, respectively. $\varepsilon_{G,\,gldas}$ is the soil permittivity calculated from the GLDAS soil simulations.

As shown in Figure 6, the snow density retrieval results with poor performance show systematic biases, with a mean bias ranging from $-77.5$ kg/m$^3$ to $108.7$ kg/m$^3$. However, although the RMSE increases to $90.9\sim131.6$ kg/m$^3$, the unbiased RMSE (ubRMSE) is within $47.5\sim74.2$ kg/m$^3$, which is comparable to the results in Figure 5. The ubRMSE can be used to remove the contribution of systematic bias to RMSE. The small ubRMSEs imply that the variation trend of the observed and the retrieved snow density is still consistent. We found that all three stations in Figure 6 are located close to water bodies, which may be frozen or unfrozen in winter depending on the air temperature and the size of the lake. Additionally, it is possible that the lake was unfrozen when the vegetation and soil roughness parameters were fitted in the snow-free season, but frozen when we were estimating the snow density. In this case, the influence of the lake on snow density retrieval is quite complex and will need to be explored in future research. We also found that the SMOS $T_B$ at the stations in Figure 6a,b is similar to that of the stations in Figure 5. However, the SMOS $T_B$ at the station in Figure 6c is much lower than that of the stations in Figure 5, for approximately 20–30 K; therefore, the retrievals were underestimated instead of overestimated. When the retrieved snow density shows positive or negative biases at the three stations near the lakes, the GLDAS snow density consistently biases low in the early and mid-snow season.

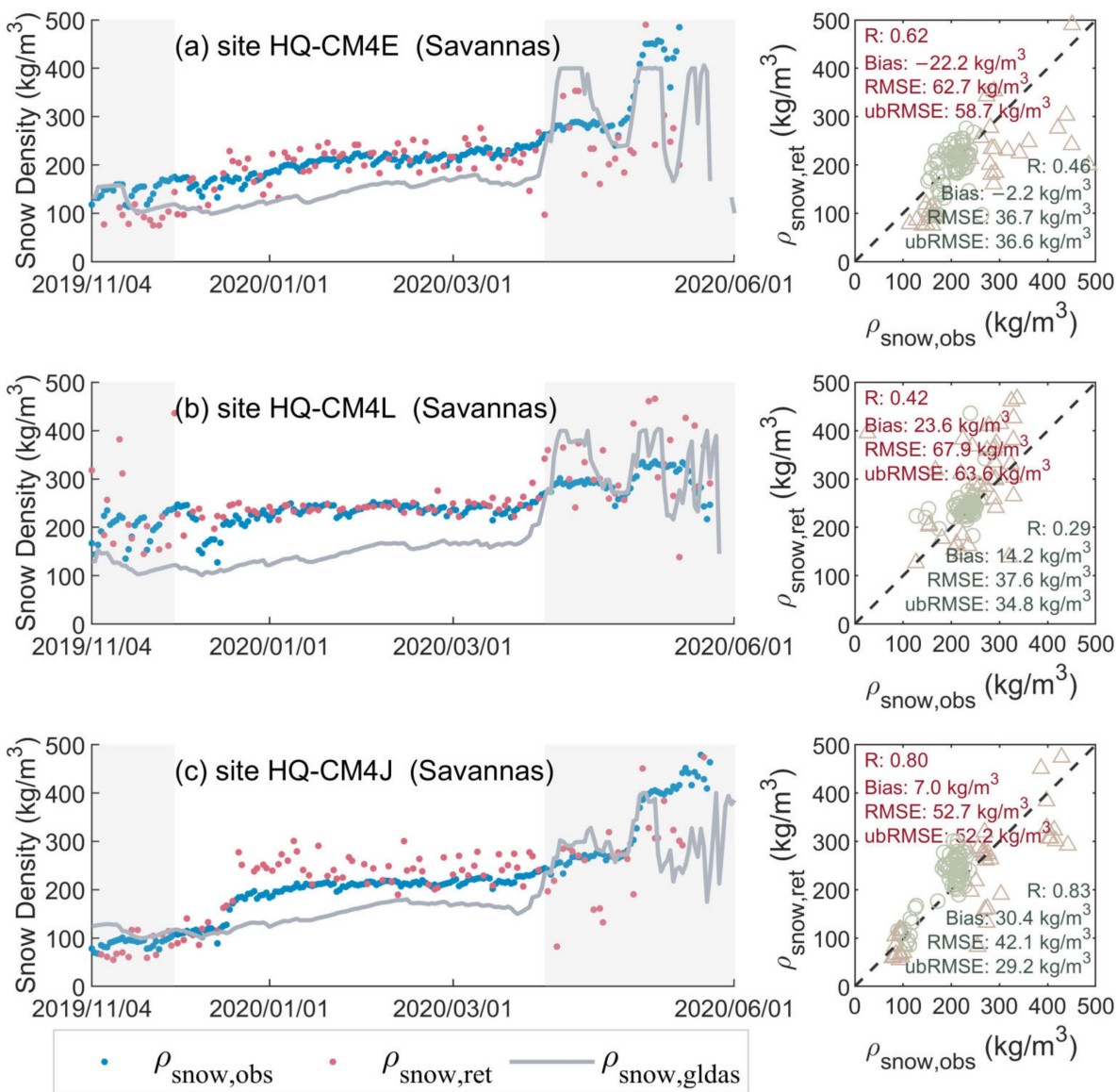

**Figure 5.** Time series (left) and scatterplots (right) of in-situ and retrieved snow density at the three stations with good performance among 43 stations: (**a**) HQ-CM4E, (**b**) HQ-CM4L, (**c**) HQ-CM4J, compared with the GLDAS snow density. In the right subplots, all points (pink triangles and green circles) and the statistics in the upper left corner present the validation in the entire snow season (October to June), whereas green circles and the statistics in the lower right corner present the validation from December to March. R is the Pearson correlation coefficient with the confidence interval of 95%, bias represents the mean bias, RMSE is the root-mean-squared error, and ubRMSE is the unbiased root-mean-squared error.

### 4.3. Validation of Retrieved Snow Density at All Stations

In Figure 7, the snow density retrieval performance is summarized from 43 stations to test the robustness of our retrieval algorithm. Figure 7a,b show the scatterplots of the SMOS-retrieved and GLDAS snow density against the measured snow density, respectively. It shows, from all validation stations, that the retrieved snow density has an RMSE of 83 kg/m$^3$ and a Pearson correlation coefficient of 0.5 with measurements, whereas the GLDAS snow density has a lower RMSE of 76 kg/m$^3$ and a higher correlation of 0.7. However, the GLDAS snow density has an overall underestimation of 48 kg/m$^3$, and the absolute bias is significantly higher than that of the retrieved snow density (9.4 kg/m$^3$). If

only the December–March period is summarized, the RMSE of the retrieved snow density improves to 72.3 kg/m$^3$, which is comparable to that of GLDAS at 71.9 kg/m$^3$. Near 200 kg/m$^3$, GLDAS underestimated the snow density, whereas our algorithm overestimated the snow density. The underestimation of GLDAS is from the early and mid-snow season, as shown in Figures 5 and 6. It implies that although the retrieved snow density contains larger noise than the reanalysis, it still has a more trustworthy temporal variation trend and a generally unbiased characteristic. Figure 7c,d show that both the retrieved and the GLDAS snow density biases are insensitive to the snow depth.

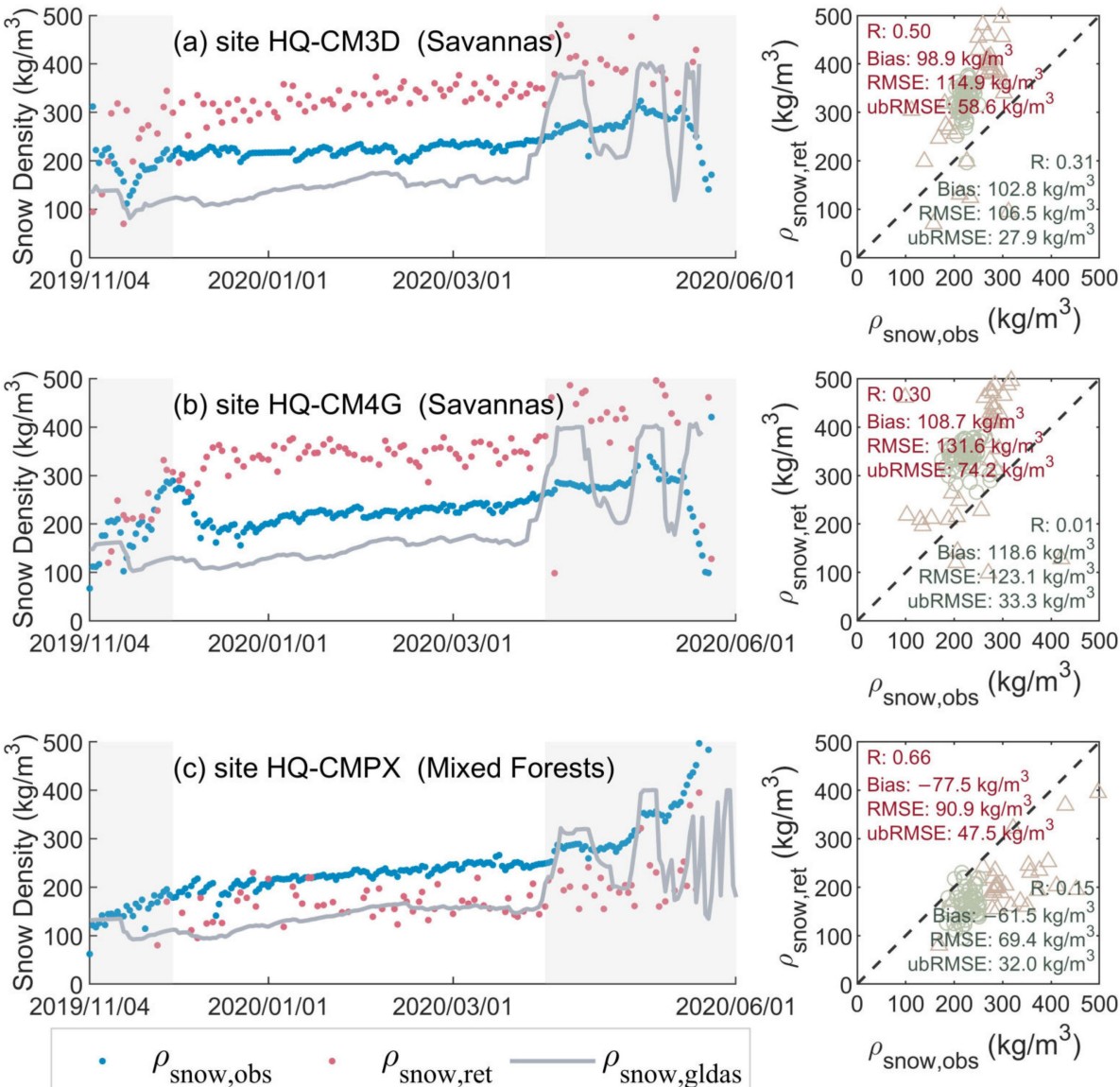

**Figure 6.** Time series (left) and scatterplots (right) of in-situ and retrieved snow density at the three stations with poor performance among 43 stations: (**a**) HQ-CM3D, (**b**) HQ-CM4G, (**c**) HQ-CMPX, compared with the GLDAS snow density. Labels and statistics in the right subplots are the same as those in Figure 5.

Figure 8 shows the distribution of validation metrics across each station on a geographic map. It shows that stations located in the savannas in the north of Quebec tend to have higher correlations with measurements and lower ubRMSEs. Conversely, stations in forests and woody savannas have lower correlation coefficients and higher ubRMSEs. However, in savannas, some stations exhibit strong overestimation and large RMSEs, with

seven out of nine located near large water bodies. Therefore, as summarized in Figure 9 and Table 1, the average values of the validation metrics for different land cover types are similar, because both forested and non-forested regions have their own challenges. More statistics on snow density retrieval accuracy at each station are provided in Appendix A.

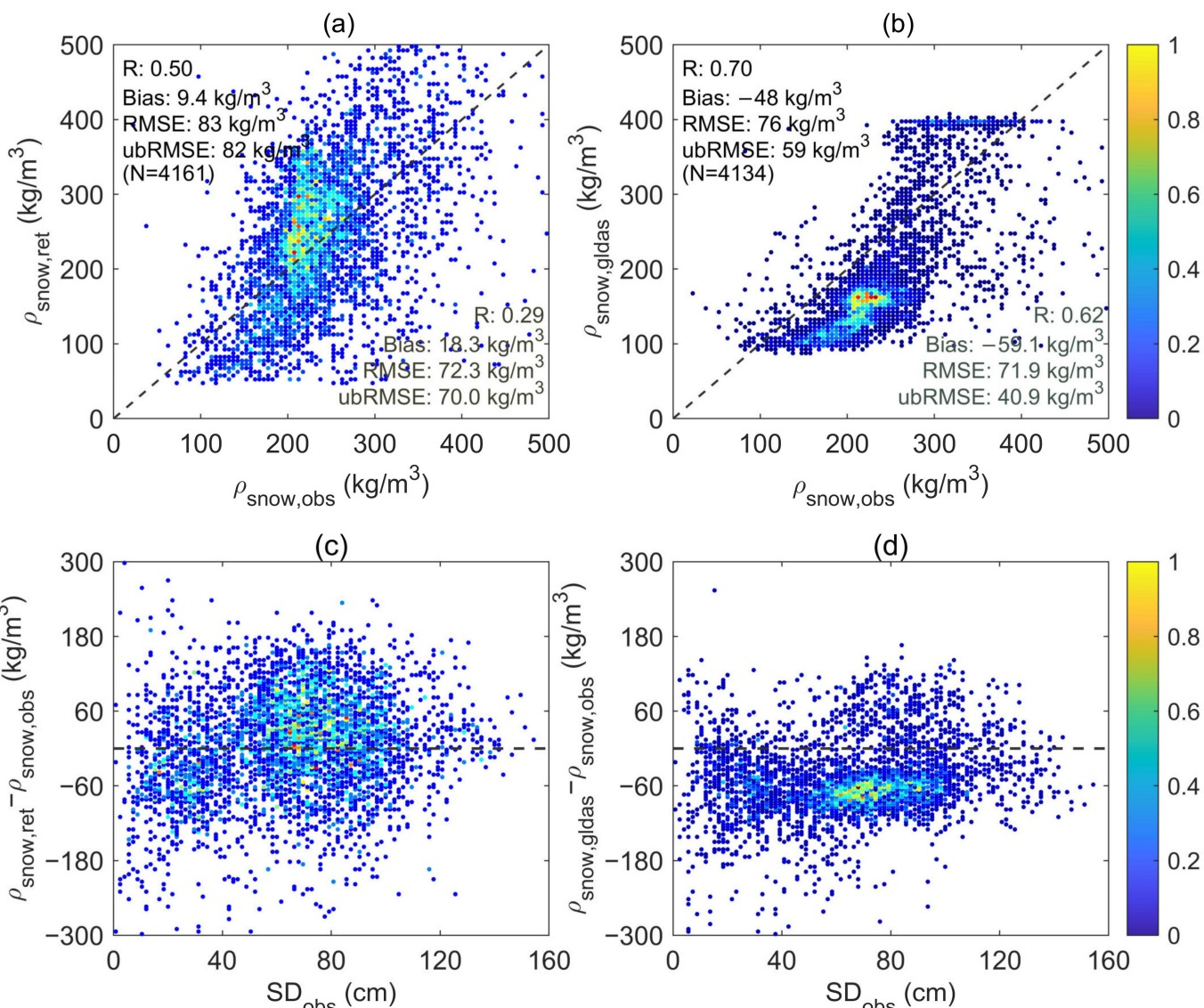

**Figure 7.** Scatterplots of (**a**) retrieved snow density and (**b**) reanalysis snow density from GLDAS against observed snow density, and (**c**,**d**) the sensitivity of biases to observed snow depth (SD$_{obs}$), from October, 2019 to June, 2020 at 43 stations located in Quebec, Canada. In (**a**,**b**), the validation metrics from the entire snow season (October to June) are presented in the upper left corner, whereas those from December to March are presented in the lower right corner.

**Table 1.** The validation metrics of retrieved snow density compared with in-situ measurements summarized based on the MCD12Q1 IGBP classification.

| MCD12Q1 IGBP Clas-sification | R | | Bias (kg/m³) | | RMSE (kg/m³) | | ubRMSE (kg/m³) | | Measurement Number | | Station Number |
|---|---|---|---|---|---|---|---|---|---|---|---|
| | October – May | December – March | October – May | December – March | October – May | December – March | October – May | December – March | October – May | December – March | |
| evergreen needleleaf forest | 0.35 | −0.19 | −33.44 | −12.95 | 78.55 | 52.97 | 71.08 | 51.36 | 93 | 62 | 1 |
| woody savannas | 0.55 | 0.39 | −16.81 | 22.27 | 85.81 | 77.07 | 84.15 | 73.78 | 1011 | 680 | 4 |
| mixed forest | 0.47 | 0.48 | 12.5 | 3.67 | 76.59 | 49.53 | 75.56 | 49.39 | 393 | 258 | 10 |
| savannas | 0.5 | 0.25 | −11.37 | 20.01 | 82.8 | 73.76 | 82.01 | 70.99 | 2664 | 1816 | 28 |
| ALL SITES | 0.5 | 0.29 | 9.44 | 18.33 | 82.89 | 72.31 | 82.35 | 69.95 | 4161 | 2816 | 43 |

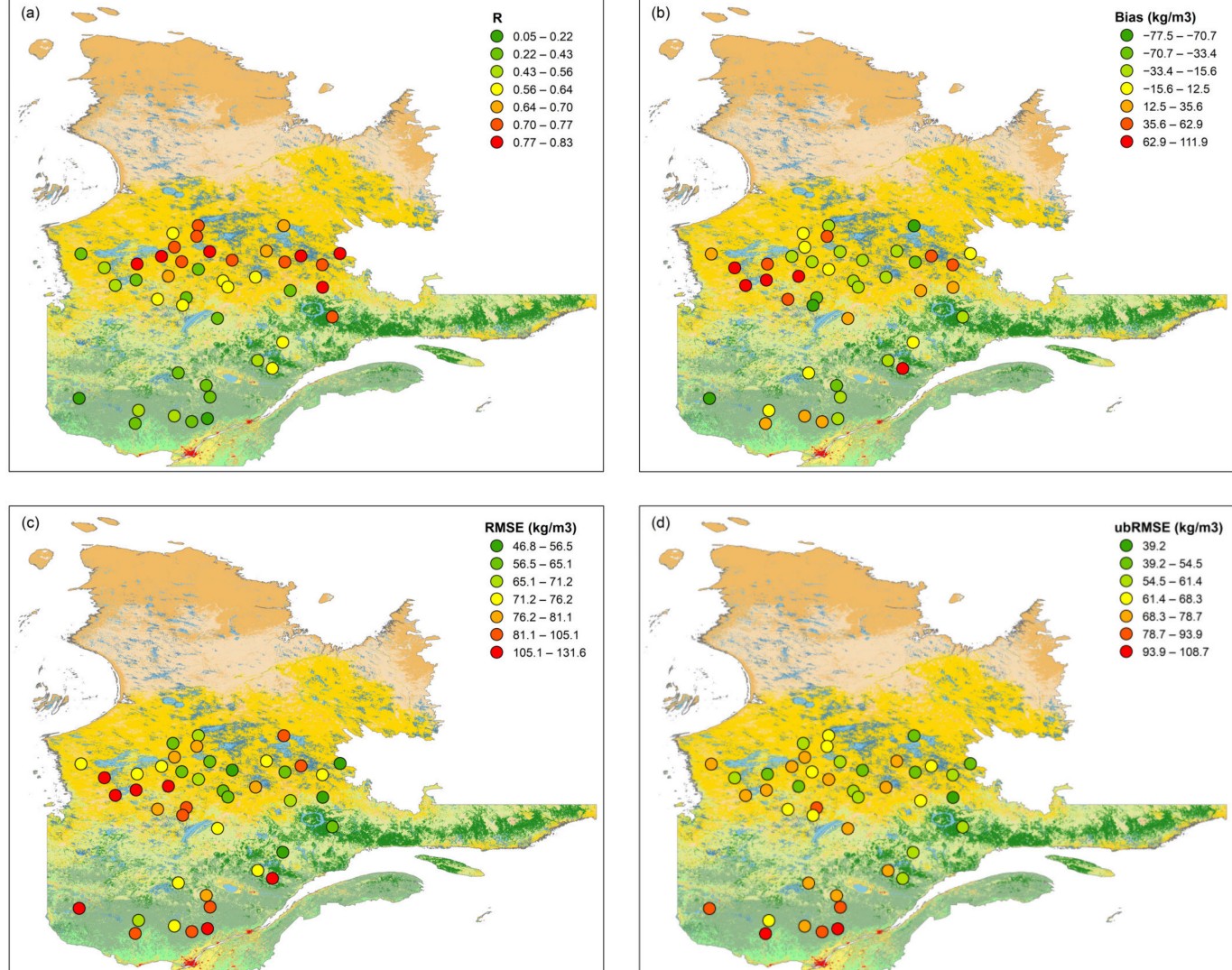

**Figure 8.** Distribution of Pearson correlation coefficient (R) (**a**), mean bias (Bias) (**b**), root-mean-squared error (RMSE) (**c**), and unbiased (ubRMSE) (**d**) of retrieved snow density at stations on map. The background is the MCD12Q1 IGBP classification.

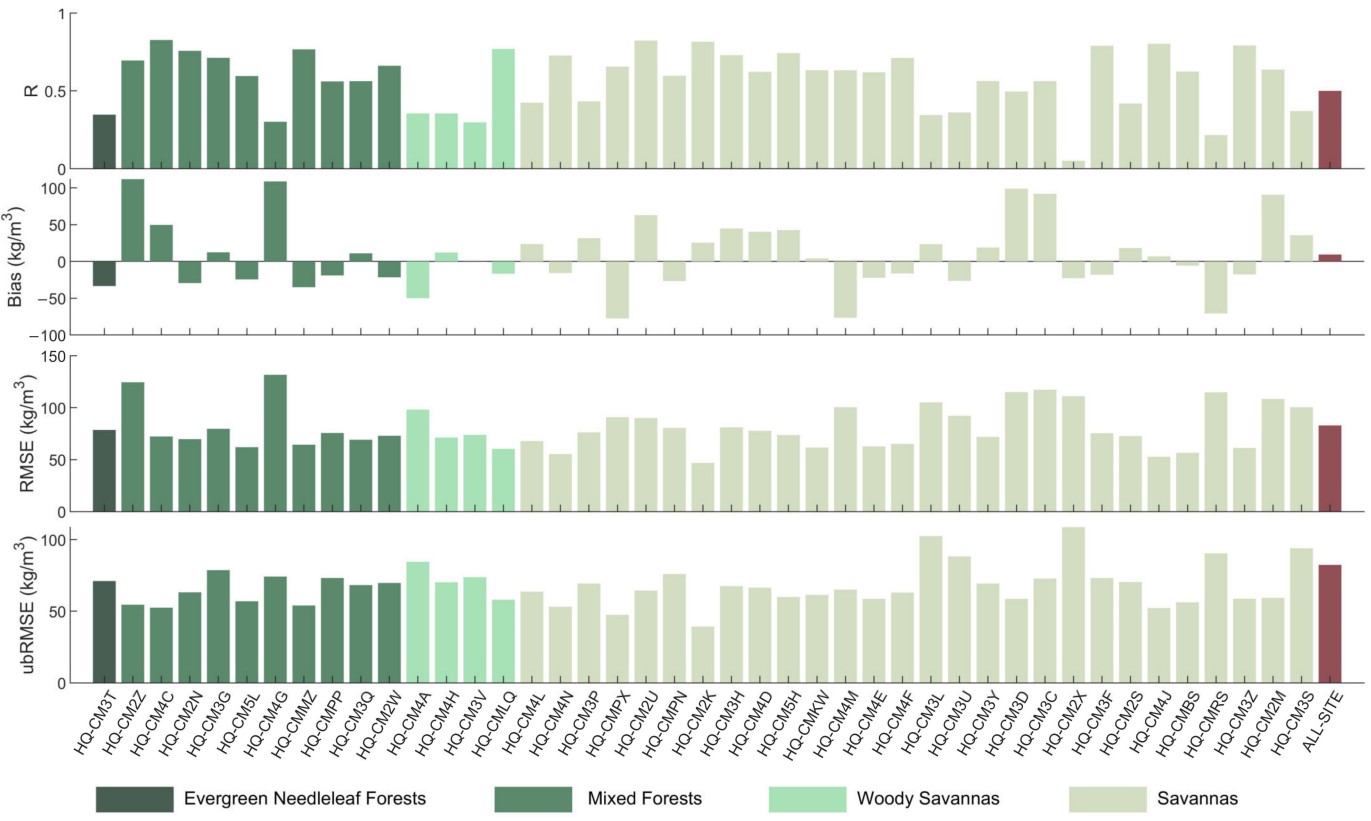

**Figure 9.** Summary of Pearson correlation coefficient (R), mean bias (Bias), root-mean-squared error (RMSE), and unbiased (ubRMSE) of retrieved snow density at different stations. Different colors represent different dominant IGBP land surface types from MCD12Q1.

## 5. Discussion

In this paper, the snow density retrieval accuracy is influenced by the following factors. Firstly, soil permittivity is a key parameter that affects the L-band multiple-angle $T_B$. However, our retrieval algorithm uses the soil permittivity calculated from the reanalysis dataset as a known variable to support snow density estimation. Secondly, the vegetation and soil roughness parameters ($\tau$, $\omega$, $S_D$) were assumed the same in snow-free and snow-covered periods. However, at least $\tau$ and $\omega$ were found to vary with the air temperature in subzero conditions, especially at low frequencies [58]. Thirdly, natural snow is a layered medium. However, we neglected the possible vertical variation in snow density in the snow profile, although the study in [7] mentioned that the retrieved snow density may be closer to the bottom snow density instead of the average. Fourthly, factors such as wet snow can make snow density retrieval impossible or unreliable. The ice lens and snow crust formed during the freeze–thaw process can complicate the snow stratigraphy, increase the refraction inside the snow, and make the current retrieval algorithm less accurate. These factors were not considered in this paper, as simplifications were made to make snow density retrievable from satellites. Additionally, the snow density retrieval may be influenced by SMOS observation errors, footprint differences in different incidence angles, lakes (frozen or unfrozen), ice, and complex landscapes, etc.

In-situ measurements used to calculate the error of our retrieval algorithm were from the point scale, with a single site inside each SMOS grid. Therefore, differences in scale between the retrieved and the measured snow density may result in an overestimation of the uncertainty of our algorithm.

In the previous section, we noted that the retrieved snow density showed stronger biases at some stations. Figure 10 shows that the systematic bias can be corrected if the three predetermined parameters ($\tau$, $\omega$, $S_D$) are artificially tuned to match the measured

snow density. However, this result assumes that the target snow density is provided. As shown in Figure 10, the bias can be reduced to below 15 kg/m$^3$ at all three stations.

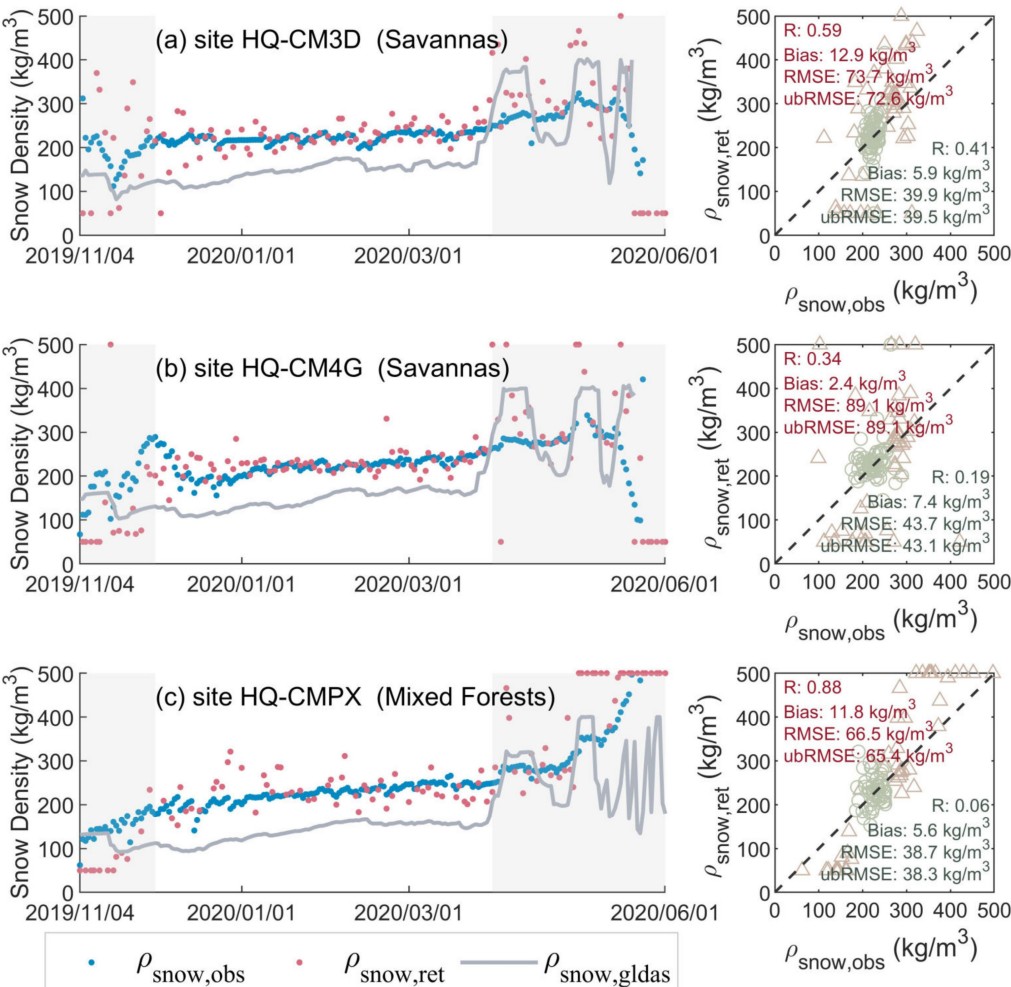

**Figure 10.** Time series and scatterplots of in-situ and retrieved snow density using manually adjusted predetermined parameters ($\tau$, $\omega$, $S_D$) at the three stations: (**a**) HQ-CM3D, (**b**) HQ-CM4G, (**c**) HQ-CMPX.

To analyze the GLDAS snow density bias compared to the measurements, we examined the relationships between GLDAS and the in-situ snow depth (SD) and snow water equivalent (SWE), as seen in Figure 11a,b, respectively. The figures reveal that many GLDAS SD values between 50 and 100 cm are unbiased but the corresponding SWEs are underestimated. Additionally, some SDs in the deep snow range are seriously underestimated. Figure 11c provides an example of one station to explain this phenomenon. GLDAS SD coincides well with in-situ measurements in the early snow season and begins to underestimate after mid-January due to the underestimated snowfall amount in GLDAS before the snowmelt. After the snowmelt onset (which occurs almost simultaneously for GLDAS and the station in mid-March), the GLDAS snowpack melts faster than the station, because it has less SWE to melt and the single-layer snow scheme utilized in the Noah model overestimates snowmelt speed [59]. Melt–refreeze events can dramatically increase the snow density. Therefore, during the snowmelt period, GLDAS overestimates the snow density compared to in-situ measurements.

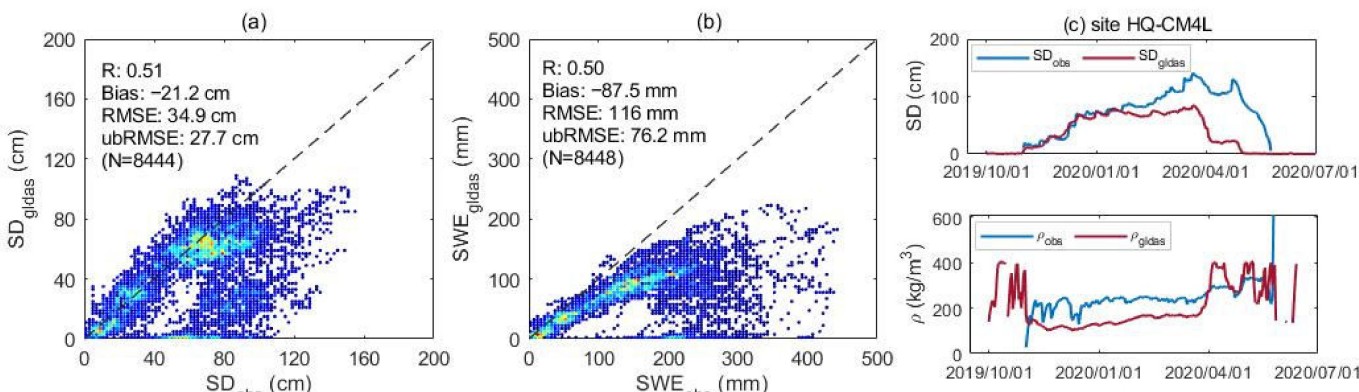

**Figure 11.** Scatterplots of (**a**) observed SD (snow depth) against reanalysis SD from GLDAS, (**b**) observed SWE against reanalysis SWE from GLDAS, and (**c**) time series of observed SD and GLDAS SD at station HQ-CM4L.

## 6. Conclusions

This study conducted snow density retrieval experiments based on L-band multiple-angle SMOS satellite observations and compared the results with the in-situ measurements from 43 CanSWE stations in Quebec, Canada. A forward model was used to describe the emission of the soil–snow–vegetation system. The vegetation and soil roughness parameters were objectively determined using SMOS $T_B$ in the snow-free period and applied to estimate the snow density. The new retrieval method achieved bias of 9.4 kg/m$^3$ and an RMSE of 83 kg/m$^3$ for snow density at all stations. Currently, some stations show large systematic biases, but these biases can be reduced.

The need for prior information about soil permittivity could be a challenge for satellite-based snow density retrieval. Similarly, to estimate the unfrozen soil water content under the snowpack, a previous study [60] introduced the snow density from the Snow Data Assimilation System (SNODAS) reanalysis dataset [61]. Whether we can develop a method to simultaneously estimate the snow density and soil permittivity from satellites, as with ground-based experiments, remains to be seen.

This paper extended the snow density retrieval from a ground-based radiometer to satellite application and established a method to solve the forest influence problem at coarse resolutions. Although the satellite-retrieved snow density is noisier than that of GLDAS, it is generally less biased and shows better time series variation characteristics. Thus, estimating the snow density from satellites using SMOS has scientific value. We anticipate a lower RMSE for SMOS-retrieved snow density if the vegetation and soil roughness parameters can be better estimated and the water body influence can be properly treated.

**Author Contributions:** Conceptualization, J.P.; methodology, J.P. and X.G.; software, J.P. and X.G.; validation, X.G.; formal analysis, J.P. and X.G.; investigation, X.G.; resources, J.P.; data curation, J.P., X.G., Z.P. and Y.B.; writing—original draft preparation, X.G.; writing—review and editing, J.P. and J.Y.; supervision, J.P., T.Z., J.Y., L.J., J.S. and L.H.; project administration, J.P. and J.S. All authors have read and agreed to the published version of the manuscript.

**Funding:** This study is supported by the National Key Research and Development Program of China (No. 2021YFB3900104), National Natural Science Foundation of China (Grant No. 41901271), Strategic Priority Research Program of the Chinese Academy of Sciences (Grant No. XDA20100300), and National Natural Science Foundation of China (Grant No. 42201346).

**Data Availability Statement:** In this study, we utilized the reprocessed SMOS dataset produced by Zhiqing Peng, whereas the SMOS raw data was obtained from SMOS L1C product (https://doi.org/10.1109/tgrs.2012.2184548 accessed on 1 June 2021). The measured snow density used for validation was from the CanSWE snow observations (https://doi.org/10.5281/zenodo.6638382 accessed on 3 December 2021). The reanalysis snow density dataset was from the GLDAS simulations (10.5067/E7TYRXPJKWOQ accessed on 18 October 2020). The forest cover and land

surface type information were from the MOD44B product (https://doi.org/10.5067/MODIS/MOD44B.006 accessed on 25 August 2020), MODIS MCD12Q1 product (https://doi.org/10.5067/MODIS/MCD12Q1.061 accessed on 7 May 2021), respectively. In addition, the soil texture auxiliary dataset was from Harmonized World Soil Database v1.2 product (FAO/IIASA/ISRIC/ISSCAS/JRC accessed on 17 February 2022).

**Acknowledgments:** The authors would like to express their gratitude to all the data providers listed in the Data Availability Statement for their dedicated efforts in supporting our study. The authors would also like to express their gratitude to Juha Lemmetyinen for providing valuable comments on the initial results of our study and highlighting the importance of the forest influence issue.

**Conflicts of Interest:** The authors declare no conflict of interest.

## Appendix A

**Table A1.** The validation metrics of retrieved snow density compared with in-situ measurements at each station.

| Stations | R | | Bias kg/m$^3$ | | RMSE kg/m$^3$ | | ubRMSE kg/m$^3$ | | Measurement Number | |
|---|---|---|---|---|---|---|---|---|---|---|
| | October – May | December – March | October – May | December – March | October – May | December – March | October – May | December – March | October – May | December – March |
| HQ-CM3T | 0.35 | - | −33.44 | −12.95 | 78.55 | 52.97 | 71.08 | 51.36 | 93 | 62 |
| HQ-CM2Z | 0.7 | 0.50 | 111.85 | 122.96 | 124.44 | 126.09 | 54.53 | 27.91 | 104 | 69 |
| HQ-CM4C | 0.83 | 0.65 | 49.63 | 51.61 | 72.24 | 64.25 | 52.49 | 38.27 | 103 | 69 |
| HQ-CM2N | 0.76 | 0.37 | −29.4 | −40.16 | 69.71 | 61.65 | 63.21 | 46.77 | 104 | 72 |
| HQ-CM3G | 0.71 | 0.57 | 12.45 | 13.95 | 79.65 | 69.60 | 78.67 | 68.18 | 107 | 73 |
| HQ-CM5L | 0.6 | 0.41 | −24.41 | −3.22 | 61.96 | 47.67 | 56.95 | 47.56 | 105 | 70 |
| HQ-CM4G | 0.3 | 0.01 | 108.75 | 118.57 | 131.65 | 123.14 | 74.19 | 33.26 | 104 | 70 |
| HQ-CMMZ | 0.77 | 0.49 | −34.98 | −31.97 | 64.34 | 50.26 | 54 | 38.78 | 109 | 72 |
| HQ-CMPP | 0.56 | 0.12 | −19.03 | 4.16 | 75.63 | 65.49 | 73.19 | 65.36 | 93 | 59 |
| HQ-CM3Q | 0.56 | 0.37 | 11 | 9.67 | 69.13 | 48.37 | 68.25 | 47.39 | 84 | 58 |
| HQ-CM2W | 0.66 | 0.27 | −21.54 | −23.67 | 72.96 | 59.21 | 69.71 | 54.28 | 101 | 68 |
| HQ-CM4A | 0.36 | 0.46 | −49.88 | −32.27 | 98.15 | 58.26 | 84.53 | 48.51 | 94 | 64 |
| HQ-CM4H | 0.36 | 0.40 | 12.14 | 31.34 | 71.24 | 51.31 | 70.19 | 40.63 | 105 | 70 |
| HQ-CM3V | 0.3 | 0.31 | −0.17 | 15.82 | 73.74 | 40.10 | 73.74 | 36.84 | 99 | 61 |
| HQ-CMLQ | 0.77 | 0.56 | −16.64 | −2.34 | 60.38 | 46.01 | 58.05 | 45.95 | 98 | 63 |
| HQ-CM4L | 0.42 | 0.29 | 23.64 | 14.16 | 67.88 | 37.56 | 63.63 | 34.79 | 102 | 60 |
| HQ-CM4N | 0.73 | 0.43 | −15.63 | −8.07 | 55.39 | 40.87 | 53.14 | 40.07 | 107 | 71 |
| HQ-CM3P | 0.43 | - | 31.69 | 52.68 | 76.22 | 75.39 | 69.32 | 53.93 | 103 | 69 |
| HQ-CMPX | 0.66 | 0.15 | −77.45 | −61.53 | 90.86 | 69.38 | 47.51 | 32.05 | 88 | 55 |
| HQ-CM2U | 0.82 | 0.55 | 62.91 | 69.30 | 90.03 | 81.75 | 64.4 | 43.36 | 109 | 71 |
| HQ-CMPN | 0.6 | 0.32 | −26.66 | −6.69 | 80.53 | 60.57 | 76 | 60.20 | 96 | 67 |
| HQ-CM2K | 0.82 | 0.65 | 25.48 | 33.71 | 46.8 | 38.72 | 39.26 | 19.05 | 105 | 64 |
| HQ-CM3H | 0.73 | 0.63 | 44.79 | 52.42 | 81.05 | 75.35 | 67.55 | 54.13 | 109 | 72 |
| HQ-CM4D | 0.62 | 0.37 | 40.4 | 49.10 | 77.74 | 71.73 | 66.41 | 52.30 | 97 | 67 |
| HQ-CM5H | 0.74 | 0.25 | 42.72 | 52.34 | 73.61 | 78.72 | 59.95 | 58.80 | 99 | 64 |
| HQ-CMKW | 0.63 | 0.58 | 4.12 | 18.10 | 61.56 | 49.20 | 61.42 | 45.75 | 115 | 73 |
| HQ-CM4M | 0.63 | 0.52 | −76.54 | −72.32 | 100.48 | 92.70 | 65.09 | 57.99 | 88 | 66 |
| HQ-CM4E | 0.62 | 0.46 | −22.16 | −2.25 | 62.7 | 36.71 | 58.65 | 36.64 | 103 | 70 |
| HQ-CM4F | 0.71 | 0.65 | −16.36 | −1.53 | 65.08 | 49.54 | 62.99 | 49.51 | 105 | 71 |
| HQ-CM3L | 0.35 | 0.08 | 23.48 | 19.26 | 105.1 | 106.59 | 102.44 | 104.84 | 53 | 49 |
| HQ-CM3U | 0.36 | - | −26.36 | −11.21 | 92.11 | 60.07 | 88.26 | 59.01 | 87 | 61 |
| HQ-CM3Y | 0.56 | 0.32 | 18.96 | 13.59 | 71.88 | 50.40 | 69.33 | 48.53 | 83 | 57 |
| HQ-CM3D | 0.5 | 0.31 | 98.86 | 102.81 | 114.94 | 106.53 | 58.64 | 27.91 | 107 | 70 |
| HQ-CM3C | 0.56 | 0.13 | 91.9 | 120.15 | 117.25 | 131.32 | 72.81 | 53.00 | 101 | 68 |
| HQ-CM2X | 0.05 | - | −22.72 | −20.42 | 111.01 | 89.50 | 108.66 | 87.14 | 79 | 57 |
| HQ-CM3F | 0.79 | 0.67 | −18.01 | −33.46 | 75.39 | 65.92 | 73.2 | 56.80 | 86 | 66 |
| HQ-CM2S | 0.42 | 0.08 | 18.14 | 34.92 | 72.66 | 59.12 | 70.36 | 47.71 | 98 | 66 |
| HQ-CM4J | 0.8 | 0.83 | 6.98 | 30.42 | 52.69 | 42.14 | 52.23 | 29.16 | 108 | 66 |
| HQ-CMBS | 0.62 | 0.21 | −5.73 | 10.30 | 56.53 | 44.99 | 56.24 | 43.79 | 100 | 63 |
| HQ-CMRS | 0.22 | - | −70.72 | −66.93 | 114.79 | 89.79 | 90.43 | 59.85 | 79 | 57 |
| HQ-CM3Z | 0.79 | 0.70 | −17.54 | −17.85 | 61.28 | 48.86 | 58.72 | 45.48 | 99 | 71 |
| HQ-CM2M | 0.64 | 0.61 | 90.63 | 106.50 | 108.32 | 113.65 | 59.33 | 39.68 | 97 | 65 |
| HQ-CM3S | 0.37 | 0.08 | 35.6 | 44.35 | 100.46 | 88.04 | 93.94 | 76.05 | 80 | 60 |
| ALL SITES | 0.5 | 0.29 | 9.44 | 18.33 | 82.89 | 72.31 | 82.35 | 69.95 | 4161 | 2816 |

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
