# Peer review of "Snow Density Retrieval in Quebec Using Space-Borne SMOS Observations"

_remotesensing, doi:10.3390/rs15082065_

Round 1
Reviewer 1 Report
Snow density has a great importance in snow observations. This paper demonstrated the snow density can be retrieved from SMOS observation, and a feasible method has been developed. The developed method is not only theoretically reasonable, but also yields reliable results, which has a significant contribution for the snow community. Therefore, we highly recommend this paper for publication with minor revisions. We offered two suggestions for the authors' consideration.
1) In equation 3, volume scattering and snow absorption are assumed to be negligible. However, factors such as wet snow, ice lens, and snow crust formed during the freeze-thaw process can increase snow scattering, making it difficult to ignore these parameters. To better demonstrate the efficacy of the snow density retrieving method developed in this paper, a relatively stable snow condition is necessary. The retrieval results are still promising, but wet snow could be considered as another types of retrieval in future studies. To mitigate the effect of the freeze-thaw process, some data collected during late Autumn (November) and early Spring (after March) could be excluded.
2) In section 4.2, we recommend distinguishing between random errors and biases, given the vast differences in observation scale between the situ stations and SMOS. The snow distribution pattern is a significant factor to consider, and therefore, the RMSE contributed by the bias should not necessarily imply poor algorithm performance.
More detailed suggestions listed:
page 2 line 53: The lower frequency channel is better suited for retrieving snow density as it is more sensitive to snow refraction rather than snow volume scattering.
page 2 line 97: To minimize the impact of forest on during the snow density retrieval, the τ - ω vegetation model is utilized.
page 3 line 102: objective method sounds awkward, maybe use a method based on optimization instead?
page 6 line 215: While it could be a promising approach to determine the value of parameters, I would hesitate to claim that it can prevent local optimization unless there is supporting evidence from credible references.
Reviewer 2 Report
The article gave us the new snow density retrieval algorithm from SMOS space-borne instruments. It is interesting and meaningful for reducing the error and representativeness of snow density data, further more reducing SWE uncertainty. But there are still some parts need to be improved and modified for refining value of this article. It is necessary to give the minor revision for this manuscript. Below are comments and suggestions.
1, For each Pearson correlation coefficient, the confidence interval should be given.
2, Paragraph 3 in Introduction: there are only one sentences for existed methods based on SAR, more should be listed.
3, Part 4.1:data time in text is different from the graph, please check again. More two days should be added, now there are only two days in two months. Personally, you should give the one day result for each month.
4, Figure 6:what is the reason of increasing gradually in figure 6a and figure 6b, but the decreasing in figure 6c? if possible, please present some discussion in later part.
5, Table 1: About expression of “Note that the statistics for evergreen needleleaf forests seems poorer because it is from one station” for table 1, it is correct? Because we know data from one station had smaller heterogeneity compared with lot of stations, in this case, the R should be better?
